# PHORECAST: ENABLING AI UNDERSTANDING OF PUBLIC HEALTH OUTREACH ACROSS POPULATIONS

## ABSTRACT

Understanding how diverse individuals and communities respond to persuasive messaging holds significant potential for advancing personalized and socially aware machine learning. While Large Vision and Language Models (VLMs) offer promise, their ability to emulate nuanced, heterogeneous human responses –particularly in high stakes domains like public health– remains underexplored due in part to the lack of comprehensive, multimodal dataset. We introduce **PHORECAST** (**P**ublic **H**ealth **O**utreach **RE**ceptivity and **CA**mpaign **S**ignal **T**racking), a multimodal dataset curated to enable fine-grained prediction of both individual-level behavioral responses and community-wide engagement patterns to health messaging. This dataset supports tasks in multimodal understanding, response prediction, personalization, and social forecasting, allowing rigorous evaluation of how well modern AI systems can emulate, interpret, and anticipate heterogeneous public sentiment and behavior. By providing a new dataset to enable AI advances for public health, PHORECAST aims to catalyze the development of models that are not only more socially aware but also aligned with the goals of adaptive and inclusive health communication.

## 1 INTRODUCTION

Predictive models of human responses to persuasive messaging are a foundational challenge in behavioral modeling, with applications spanning social science, policy, and AI alignment. A key obstacle is simulating how individuals with diverse demographics, personalities, and cultural backgrounds, react to the same stimulus (e.g., an image, text, or video). While vision-language models (VLMs) offer a potential solution, it is not clear how well calibrated these models are, or how well they simulate differences between demographic groups. This gap stems from a misalignment between standard VLM training objectives (e.g., benchmark accuracy Xiao et al. (2025)) and the nuanced demands of behavioral simulation - a task requiring fine-grained preference elicitation and demographic-aware calibration Squires et al. (2023); Langellier (2016). To address this, we argue for domain-specific tuning of VLMs using human response data, which we demonstrate through the lens of public health messaging, a high-impact domain where tailored messaging can effectively promote awareness, shift attitudes, and inspire healthier behaviors at scale (Adegoke et al., 2024; Conway & Others, 2025; Ghio et al., 2021).

Despite advances in behavioral science, there remains no comprehensive dataset capturing how individuals—across diverse demographic backgrounds and personality profiles—respond to real-world health messages. To address this gap and catalyze research on understanding individual public health preferences, we introduce a novel dataset derived from a large-scale study of over 1,000 participants. This dataset comprises of **30,000+** rich, granular responses to 37 public health posters spanning 7 urgent health topics (e.g., COPD, Mental Health, Nutrition, and more). Each response reflects sentiment, emotional reactions, and behavioral intent, offering unprecedented insight into the interplay between messaging design, individual differences, and community-level engagement. By pairing these responses with detailed demographic and psychometric data, we empower researchers to build predictive models of how public health campaigns are perceived by different groups.

We provide insights to study how unique individuals interact with and react to various multi-media marketing content. We analyze and present the correlation between different demographic factors and personality traits, as well as with their individual responses to varying public health messaging.

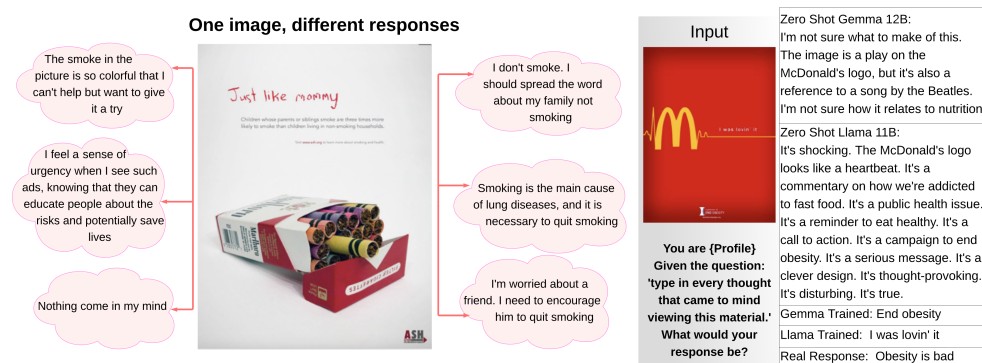

Figure 1: Human Nuance vs Model Limitation: Qualitative analysis demonstrates the diverse human reactions and interpretations evoked by a single image, spanning personal concern to broader advocacy. This underscore the significant influence an individual's background and context has on their perception. The right panel shows that current popular models struggle to capture this rich spectrum of human responses and language, often defaulting to repetitive language (e.g., Llama's "I'm not sure..." in 80% of cases). By training with PHORECAST, our models learn to emulate real human language, effectively capturing these subtle distinctions in the data.

We demonstrate the utility of this dataset through two important use cases: (1) training predictive models to simulate response to public health messaging based on demographic and psychographic factors, and (2) establishing the first benchmark for evaluating how personality traits modulate emotional responsiveness to visual persuasion. PHORECAST enables LLM models to better align with individual preferences and values, given their demographics and/or personality traits.

## 2 RELATED WORK

**Simulating Human Behavior with Language Models.** A lot of recent work explores the idea of using large language models (LLMs) as simulators for human behavior. Park et al. Park et al. (2023) is one of the first works to investigate emergent human interaction behavior by simulating a sandbox human community with multiple LLM instances. This inspired many branch-off topics involving LLM agents, especially for human behavior simulation, a popular and well-motivated area, particularly for healthcare studies or for commerce platform optimization Lu et al. (2025). One large motivator for human behavior simulation research is the prospect of being able to simulate large-scale social media populations to major political events or digital campaigns. Qiu et al. Qiu et al. (2025) investigates the scope of simulating social media behavior through action-conditioned free text responses, where the actions can be either "like", "reply", or "quote". The human data scraped from X revolved around major political events. They found that baseline GPT and Deepseek models are biased heavily towards selecting "quote" over other actions, which may suggest that complementary text is preferred over direct replies or text-free likes. Another study on social media simulation Li et al. (2024) found that historical context was by-far the most important information for accurate simulation of human responses, compared to user interests and user info (such as demographics). Unlike previous work, Xie et al. Xie et al. (2025) built a cognitive science-inspired framework for the simulation of detailed human backgrounds, offering a much more explicit and robust way to construct simulated human personas. Instead of other works using personality tests like the Meyers-Briggs test Song et al. (2024), Xie et al. construct the first framework that uses Jung's psychology theory. In contrast to previous work, our dataset introduces a domain-specific behavior-dependent prediction in the public health domain. The responses include not only free responses, but also self-reported personality evaluations according to the Big Five Inventory John et al. (1991), as well as detailed demographic information.

## 3 THE PHORECAST DATASET

PHORECAST (**P**ublic **H**ealth **O**utreach **RE**ceptivity and **CA**mpaign **S**ignal **T**racking) is a multimodal dataset designed to evaluate how vision-language models (VLMs) predict human reactions to public health campaigns, conditioned on demographic and psychological factors. It comprises survey

**Unique Demographics in PHORECAST**

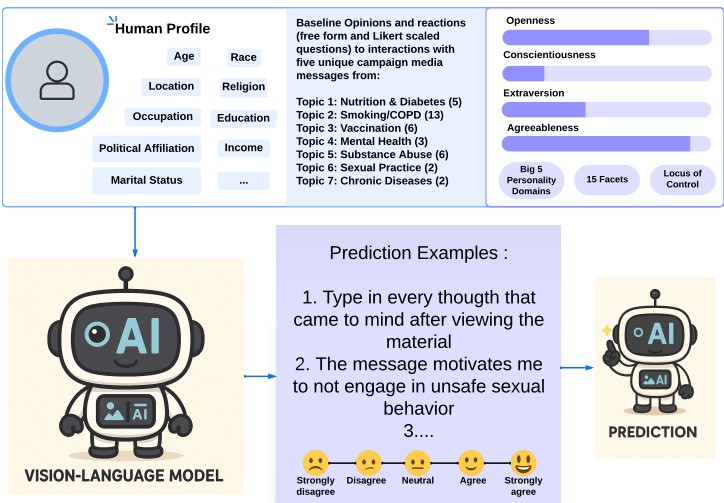

Figure 2: Demographic distribution of participants (N=1095) in our study, showing age groups, gender identity, religion, political affiliation, and highest education attainment. The dataset reflects a broad representation across ages (predominantly 18-44), gender (balanced male/female, inclusive non-binary options), and political views (moderate, liberal, and conservative as most frequent).

responses from diverse U.S. participants, linking structured annotations of health media to rich individual profiles. We recruit participants (age $\geq$ 18, U.S. residents) who provide informed consent and complete a 30-minute anonymized survey. Duplicate IP addresses are filtered to ensure uniqueness. Each participant (1) Profiles their Background: Reports demographics, personality traits, the locus of control and baseline health opinions on five randomly selected topics (Section 3.1), (2) Reviews Campaigns: Reacts to five randomly assigned public health campaigns (based on the randomly selected topics) via free-form text and Likert-scale ratings (Section 3.2), as shown in Fig. 3.

Figure 3: **Overview of PHORECAST Pipeline**. Via our Survey, we collect human profiles including demographics, personality, locus of control, and opinions on public health topics before and after their interaction with the campaign message. We then train LLM/VLM models to predict different reactions of an individual given a stimuli.

### 3.1 SURVEY DETAILS

**Health Topics:** Public health experts from our team curate campaigns from the web and annotate with target behavior (e.g., smoking cessation), target population, and message type (Informative, Persuasive-Efficacy, or Persuasive-Threat). Each participant is assigned five random topics at the start of the survey from seven categories: Nutrition & Diabetes, Vaccination/HIV/AIDs, Mental Health, Substance Abuse, Sexual Practices, COPD/Smoking, and Chronic Diseases (which includes Heart Disease, Cystic Fibrosis, and Arthritis).

**Basic demographics:** We request the following demographics features from each participant: age, gender, assigned sex at birth, religious or cultural affiliation, political affiliation, race, ethnicity, primary & first language, educational attainment, employment status, current profession, annual household income, marital status, family status, physical or health conditions, and zip code. A "Prefer not to say" option is offered for each demographic question. For gender, participants can select *Male*, *Female*, *Non-Binary/Third Gender*, or *self-identify*. We collect self-described ethnicity, religion and

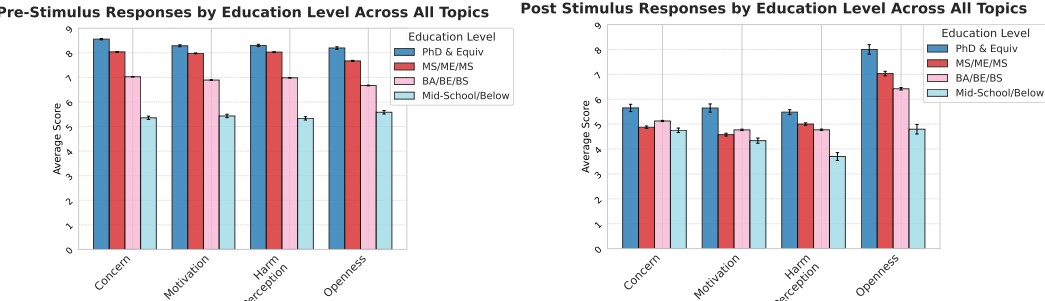

Figure 4: Differential opinion patterns by education level before and after interacting with stimuli across all topics. Generally, individuals with higher education attainment (Doctoral and Masters; N=183) demonstrate (1) significantly greater concern about different aspects of their health, (2) stronger harm perception of health harms, (3), paradoxically, greater self-reported willingness to engage in harmful behavior such as substance use or smoking. This attitude behavior gap suggests that while higher education enhances risk awareness, it may simultaneously increase behavioral intentions, possibly through increase perceived self behavioral control Assari et al. (2019). A detailed demographic and psychographic analysis for each topic is provided in the appendix.

political affiliations, employment, marital and family status, as well as first language. This set of demographic information captures the subtle differences in background and current factors that may influence an individual's perception and/or opinions on diverse issues.

**Personality:** Each participant completes The Big Five Inventory-2 (BFI-2) Soto & John (2017), a 60-item questionnaire designed to measure the Big Five personality domains (Extraversion, Agreeableness, Conscientiousness, Negative Emotionality, and Open-Mindedness) and their 15 facets: sociability, assertiveness, energy, compassion, respectful, trust, organization, productive, responsibility, anxiety, depression, emotional volatility, intellectual curiosity, aesthetic sensibility and creative imagination.

**Locus of Control:** The locus of control measures one's tendency to perceive events as internally or externally controlled. Prior research links an internal locus of control to greater psychological well-being Shojaee & French (2014). Participants rate four statements ("*I'm my own boss*", "If *I work hard, I succeed*", "W*hat I do is determined by others*", "*Fate often gets in the way of my plans*") on a five-point scale ("*Does not apply at all*" to "*Applies completely*"). Total scores (4-20) categorize respondents into: Low internal (high external) locus (4–11), moderate balance (12-15), and high internal locus (16-20).

**Baseline Opinions:** Each participant rate pre-existing concern, motivation, harm perception, and openness, on a 9-point Likert scale (1 = Not at all, 9 = Extremely) on the five topics randomly assigned at the start of the survey. Concern reflects worry about risks from unhealthy behaviors (e.g. poor diet), motivation captures intent to adopt healthy behaviors (e.g., vaccinations), harm perception assesses beliefs about the consequences of neglecting safe practices, and openness gauges willingness to engage in health-supportive behaviors. For inherently harmful topics (e.g., smoking), openness instead reflects receptivity to those behaviors.

## 3.2 OPINION INDICATORS

We assess participants' reactions to health campaigns through structured and open-ended measures. Emotional responses are quantified via eight discrete emotions (sadness, anger, fear, guilt, disgust, worry, shame, and hope), rated on a 9-point Likert scale (1 = Not at all, 9 = Extremely). We reevaluate the four baseline constructs –concern, motivation, harm perception, and openness– but now specifically framed by the campaign content: (1) concern reflects message-induced worry about health risks, (2) motivation captures behavior change intent, (3) harm perception assesses consequences of non-compliance, and (4) openness measures receptivity to recommendation (reverse coded for harmful behaviors like smoking). Finally, participants provide free-form responses to '*type every thought that came to mind when viewing this material*', yielding qualitative data that complements quantitative ratings.

# 4 TRAINING AND BENCHMARKING LLM/VLM MODELS USING PHORECAST

Using PHORECAST, we train and evaluate two state-of-the-art VLMs: Llama 3.2-Vision 11B and Gemma 3 12B. Our baseline results (Table 1) reveal Llama-11B achieving 16.82% exact-match accuracy (60.66% at $\pm 2$ off), while Gemma-12B exhibits marginally better baseline performance at 23.89% exact match (78.26% at $\pm 2$ off). Both models struggle particularly with emotion and openness, suggesting these dimensions present greater challenges for current architectures. The consistent performance gap between exact and approximate matching underscores the inherent difficulty in precisely modeling human psychological responses.

| Model | Concern | Emotion | Harm Perception | Motivation | Openness | Total Average |
|---|---|---|---|---|---|---|
| Random Baseline | 14.29 / 54.86 / 68.54 | 17.69 / 30.42 / 41.66 | 21.73 / **77.96** / **88.75** | 14.13 / 50.00 / 62.92 | 14.59 / 47.57 / 66.11 | 16.49 / 52.16 / 65.60 |
| **Llama 11B** | | | | | | |
| Base | 16.57 / 48.18 / 62.46 | 14.03 / 32.55 / 45.53 | 21.88 / 66.57 / 78.88 | 15.81 / 46.35 / 60.03 | 15.81 / 41.03 / 56.38 | 16.82 / 46.94 / 60.66 |
| Trained with PHORECAST | **42.71** / **72.49** / **87.69** | 37.39 / **56.98** / **70.94** | **43.16** / 64.89 / 85.41 | 40.88 / **67.63** / **83.89** | **36.47** / **66.26** / **83.28** | **40.12** / **65.65** / **82.24** |
| **Gemma 12B** | | | | | | |
| Base | 26.44 / 63.22 / 86.78 | 13.97 / 40.88 / 60.30 | 29.33 / 71.43 / 88.45 | 24.77 / 60.18 / 79.94 | 24.92 / 55.62 / 75.84 | 23.89 / 58.27 / 78.26 |
| Trained with PHORECAST | 42.40 / 67.93 / 84.80 | **37.55** / 54.35 / 69.45 | 42.25 / 65.20 / 81.76 | **41.64** / 64.13 / 81.91 | 36.32 / 61.25 / 80.24 | 40.03 / 62.57 / 79.63 |

Table 1: **Evaluation Accuracy Using Partial Profiles:** Model accuracy across five response dimensions using partial profiles before and after training: this case tests the most applicable real use case, where we may only have a partial profile of an individual we wish to predict an opinion for. We use the procedure described above to create a split consisting of partial features (personality, demographics, and in-context Q/As), with the stratified strategy, creating over 8k evaluation samples of 521 unique individuals. Each cell reports accuracy at exact / $\pm 1$ / $\pm 2$ tolerance levels. With $\pm 2$ tolerance threshold, Llama Base achieves 45% on emotion prediction, which are measured based on 8 distinct emotional predictions. After training with our dataset, Llama 11B reaches 70.94% accuracy in the emotion category, contributing to its superior overall performance of 82.24% across all question categories, outperforming Gemma 12B. These results showcase a notable **2.4×** *increase in exact-match accuracy improvement using PHORECAST for training*.

We employ a stratified hold-out strategy across gender, religion, and race/ethnicity demographics, as described in detail in the supplementary. This approach allows for the assessment of model performance on underrepresented groups and the quantification of any systematic biases in response predictions, shown and discussed in 2. Additionally, a representative image for each topic is excluded from the training and included in the validation set to evaluate models' generalization capabilities to novel campaign visuals across all public health campaigns. We employ Low-Rank Adaptation (LoRA) for efficient fine-tuning of models, optimizing two parallel objectives: a) Free-form natural language response generation, b) Likert-scale opinion prediction. To enhance robustness for real-world deployment scenarios with partial user data, we implement feature randomization protocol during training: (1) Demographic/Psychographic Features: 90% random sampling per participant (2) Locus of Control: 75% random sampling per participant, (3) Contextual Q/A pairs: 30% sampling of question-answer pairs with variable feature subsets (randomly selected combinations of available features). This stochastic regime forces models to operate under partial information, simulating real world constraints where complete user profiles are unavailable.

## 4.1 PREDICTING OPINION INDICATORS

We assess performance through both exact and approximate matching ($\pm 2$) of predicted numerical responses across five psychological dimensions: emotion, concern, openness, motivation, and harm perception, providing comprehensive and category-specific assessments.

## 4.2 BENCHMARKING PERFORMANCE ON FREEFORM RESPONSES

Above, we measure model performance on discrete opinion scales, making accuracy easy to quantify. Measuring the accuracy of freeform text responses is more complex. We don't expect a model to perfectly match a participant's freeform responses, but rather we hope for model outputs to be distributionally similar to humans, conditioned on their demographics/personality.

To this end, we use multiple evaluations approaches: **Semantic Text Similarity (STS)** to measure how closely model responses align with human answers in meaning, **Statistical Distribution of Embeddings (SDE)** –stratified by personality traits to assess distributional alignment across different

Table 2: **Demographic Analysis Before and After Training.** We prompt the model with personality and demographics information and calculate the accuracy across subgroups, revealing demographic disparities and changes after training using $\pm 2$ tolerance threshold. We compare Gemma and Llama pre- and post-training. Notable trends: (1) accuracy improves overall, with Llama benefiting much more (over $17\%$) increase on average, (2) disparities persist for some groups, especially underrepresented samples, such as agnostics (with only 24 samples) and non-binary (36 samples).

| Demographic Group | Subgroup (Samples) | Gemma (Pre → Post) | Gemma Δ | Llama (Pre → Post) | Llama Δ |
|---|---|---|---|---|---|
| **Age** | 18–24 (994) | 65.39% → 63.48% | -1.91 | 48.09% → 65.59% | +17.50 |
| | 25–34 (3867) | 68.37% → 64.70% | -3.67 | 55.29% → 65.62% | +10.33 |
| | 35–44 (2419) | 66.76% → 71.64% | +4.88 | 49.81% → 73.97% | +24.16 |
| | 45–54 (479) | 66.81% → 82.05% | +15.24 | 51.98% → 85.62% | +33.64 |
| | 55–64 (108) | 63.89% → 86.11% | +22.22 | 50.00% → 86.11% | +36.11 |
| | 65+ (12) | 50.00% → 66.67% | +16.67 | 66.67% → 58.33% | -8.34 |
| **Gender** | Male (4081) | 67.36% → 67.51% | +0.15 | 66.80% → 69.92% | +3.12 |
| | Female (3738) | 67.07% → 68.65% | +1.58 | 52.17% → 69.54% | +17.37 |
| | Non-Binary (36) | 88.89% → 58.33% | -30.56 | 72.22% → 52.78% | -19.44 |
| | Prefer Not to Say (24) | 66.67% → 52.60% | -14.07 | 50.00% → 53.65% | +3.65 |
| **Religion** | Christianity (5936) | 64.60% → 67.05% | +2.45 | 53.82% → 69.34% | +15.52 |
| | Judaism (131) | 52.67% → 58.02% | +5.35 | 41.98% → 62.60% | +20.62 |
| | Islam (612) | 62.91% → 67.65% | +4.74 | 50.98% → 68.84% | +17.86 |
| | Buddhism (192) | 73.96% → 75.52% | +1.56 | 49.48% → 69.79% | +20.31 |
| | Hinduism (48) | 60.42% → 62.50% | +2.08 | 58.33% → 47.92% | -10.41 |
| | Agnostic (24) | 91.67% → 62.50% | -29.17 | 70.83% → 54.17% | -16.66 |
| | None (744) | 65.32% → 80.51% | +15.19 | 45.83% → 80.13% | +34.30 |
| | Prefer Not to Say (192) | 61.46% → 52.60% | -8.86 | 46.35% → 53.65% | +7.30 |
| **Overall** | All (7879) | 66.92% → 68.02% | +1.10 | 52.44% → 69.66% | +17.22 |

*Sample counts in parentheses.* **Bold** *indicates lowest accuracy in subgroup.* Underlined *Δ values show top 3 improvements per model. Overall shows weighted average accuracy across all samples in the validation set.*

subgroups, and a **discriminator-based** accuracy metric to determine how well a model can distinguish between human and machine-generated responses. We focus on semantic similarly and refer readers to the appendix for additional measures of distributional fidelity in the Supplementary Materials.

We compute the similarity (0-1 score) between each machine generated response and its expected (ground truth) response using *all-mpnet-base-v2*. We evaluate the semantic similarity scores for different subgroups to analyze which individuals or groups are *easier* to emulate. Table 3 shows the results from benchmarking Gemma and Llama across different demographic groups and the Big 5. As seen, both models benefit from our dataset. In particular, our ability to emulate females, individuals aged 45-54, and Muslims, greatly enhances after training. Interestingly, Asians, a group underrepresented in our study, remain difficult to emulate.

### 4.3 ABLATION STUDY ON GROUPS OF ATTRIBUTES

We investigate the feature importance of demographics, personality, locus of control and in context Q/As in predicting the individual's opinion using our best-trained model. These experiments aim to analyze model performance degradation and identify which features contribute most to the model's ability to emulate human response. To assess feature contribution, we prompt LLama (trained on randomized partial profiles) with only one feature group at a time – demographics, personality traits, locus of control, or in-context Q/As. We employ target masking to isolate specific features: For demographics, we exclude gender, race/ethnicity, education, or religion. For personality, we mask either the locus of control, the Big Five traits, or both. Finally, random masking removes $k$ random features from a specified group (demographics, personality, or in-context Q/As) before inference. Some key observations include:
(1) Providing the model with *personality information achieves higher accuracy (*$77.71\%$*) than providing the model with only demographics features (*$74.89\%$*), (2) providing the locus of control along with the personality is better than providing only personality or just locus. (3) In-context opinion indicators, as opposed to demographics and personality, are sufficient to predict an aspect of that individual's opinion*. In particular, providing the model with the individual's responses to the stimuli (e.g., harm perception, concern), without the free form or the initial opinions pre-stimuli, is sufficient to achieve the highest score of $95\%$ (w/ $\pm 2$) and $78\%$ at exact.

Table 3: Analysis of Model Response Similarity Across Demographic and Personality Groups: We present the response similarity scores (range: 0-1) for Gemma and Llama models, segmented by key traits. **Bold** indicates the highest score per group; underlined indicates the lowest. Post-training, both models show improved alignment, with Gemma's similarity improving from 0.32 to 0.37 and Llama from 0.28 to 0.34.

| Group | Subcategory | Gemma 12B | | Llama 11B | | Max Δ |
|---|---|---|---|---|---|---|
| | | Before | After | Before | After | |
| **Age** | 18–24 (994) | 0.30 | 0.31 | 0.27 | 0.34 | +0.07 |
| | 25–34 (3867) | 0.31 | 0.33 | 0.27 | 0.33 | +0.06 |
| | 35–44 (2419) | 0.30 | 0.33 | 0.29 | 0.32 | +0.03 |
| | 45–54 (479) | **0.41** | **0.47** | **0.37** | **0.42** | +0.06 |
| | 55–64 (108) | 0.31 | 0.37 | 0.29 | 0.29 | +0.06 |
| | **Avg.** | 0.31 | 0.34 | 0.29 | 0.33 | +0.04 |
| **Gender** | Male (4081) | 0.33 | 0.34 | 0.29 | 0.34 | +0.05 |
| | Female (3738) | 0.30 | 0.33 | 0.28 | 0.33 | +0.05 |
| | **Avg.** | 0.31 | 0.33 | 0.28 | 0.33 | +0.05 |
| **Religion** | Christianity (5936) | 0.31 | 0.34 | 0.29 | 0.33 | +0.04 |
| | Islam (612) | **0.36** | **0.42** | 0.29 | **0.41** | **+0.12** |
| | Judaism (131) | 0.36 | 0.36 | 0.32 | 0.29 | +0.00 |
| | Buddhism (192) | 0.38 | 0.34 | 0.28 | 0.31 | +0.03 |
| | Other (936) | 0.32 | 0.30 | 0.25 | 0.37 | **+0.12** |
| | **Avg.** | 0.34 | 0.35 | 0.28 | 0.34 | +0.06 |
| **Race** | White (434) | 0.31 | 0.33 | 0.28 | 0.33 | +0.05 |
| | Black (177) | 0.33 | 0.35 | 0.30 | **0.37** | +0.07 |
| | Hispanic (24) | 0.33 | 0.30 | 0.32 | 0.36 | +0.04 |
| | Asian (14) | 0.14 | 0.19 | 0.18 | 0.22 | +0.05 |
| | Other (481) | **0.34** | **0.37** | 0.29 | 0.36 | **+0.11** |
| | **Avg.** | 0.29 | 0.31 | 0.27 | 0.33 | +0.06 |
| **Big 5** | Extraversion (658) | 0.31 | **0.35** | **0.33** | 0.34 | +0.04 |
| | Agreeableness (658) | 0.31 | 0.33 | 0.29 | 0.34 | +0.05 |
| | Conscientiousness (658) | 0.31 | 0.34 | 0.29 | 0.29 | +0.02 |
| | Neuroticism (658) | 0.32 | 0.34 | 0.28 | 0.34 | +0.06 |
| | Openness (658) | **0.32** | 0.34 | 0.29 | **0.34** | +0.05 |
| | **Avg.** | 0.31 | 0.34 | 0.30 | 0.33 | +0.04 |
| **Overall Avg.** | | **0.32** | **0.37** | **0.28** | **0.34** | **+0.06** |

### 4.3.1 DEMOGRAPHICS

We evaluate the impact demographics has on the model's ability to emulate different individuals (Table 4). We begin by prompting our model with only the demographics information, and establish a baseline of $30.64\%$ (exact) and $-74.80\%$ (w/ $\pm2$) across all question categories. Next, we investigate the effect of different demographic features such as age, gender and race. We observe that prompting the model with all the demographics information produces higher accuracy than when some of those demographic features are masked. However, not at all demographics are as important: When we randomly mask 7 fields, our accuracy drops only by approximately 2%.

### 4.3.2 PERSONALITY AND LOCUS OF CONTROL

In the second experiment , we investigate how personality traits influence predictive performance (Table 5). We operationalize personality through two complementary frameworks: the BFI-2 and the locus of control (LOC). We evaluate the model using: 1. Personality alone, 2. LOC alone, 3. Big 5 domains alone, 4. Big Five + LOC, 5. Full psychological profile (personality + LOC). We replicate the random masking procedure from experiment 1 to assess stability and quantify information loss when partial trait data is available. As seen with the random masking, our model's performance does not degrade, suggesting that partial personality information can be sufficient for the model to predict an individual's opinion.

### 4.3.3 IN-CONTEXT QUESTIONS AND ANSWERS

In context questions and answers provide the model hints on how the participant responded to other questions asked. For example, if the model is trying to predict the participant's harm perception, then we ask, is providing the emotional responses to that topic helpful? We evaluate the impact of in-context information on model accuracy (Table 6). When we provide the model only with the in-context demonstrations of their reactions to the stimuli, it becomes trivial for the model to complete

Table 4: **Relative Impact of Demographic Factors on Prediction Accuracy**: We quantify the relative importance of demographic factors (gender, race/ethnicity, education, religion) in predicting responses to public health messages using our trained Llama 11B. Our study reveals: (1) education removal causes the largest performance drop $(-1.08/-1.29)$ in average $\pm 1/2$ accuracy. (2) gender and race show minimal impact ($\pm 0.63/\pm 0.15$ $\Delta$), and (3) models degrade gracefully with random feature removal ($\leq 2.21$ drop with 7 fields masked). Results demonstrate that *education-level information is the most critical for accurate prediction, while other demographics contribute modestly.*

| Comparison | Concern | Emotion | Harm | Motivation | Openness | Avg. |
|---|---|---|---|---|---|---|
| **Reference Model** | | | *All Demographics (Baseline)* | | | |
| | 30.55 / 58.21 / 81.61 | 21.90 / 44.43 / 58.78 | 39.67 / 61.85 / 84.80 | 28.42 / 54.10 / 74.62 | 32.67 / 57.90 / 74.62 | 30.64 / 55.30 / 74.89 |
| *Single Demographic Removals ($\Delta$ from Reference)* | | | | | | |
| - Gender | +0.45 / +1.21 / +0.30 | -0.21 / -0.40 / -0.12 | -0.16 / -0.45 / -0.61 | -1.67 / -2.12 / +0.76 | -0.60 / -1.37 / -1.06 | -0.44 / -0.63 / -0.15 |
| - Race | +0.61 / +0.76 / +0.61 | +0.23 / +0.30 / -0.10 | +0.15 / -0.60 / -0.76 | +0.30 / +1.22 / -0.00 | -2.88 / -2.28 / -0.00 | -0.32 / -0.12 / -0.05 |
| - Education | -0.31 / -0.00 / -1.97 | -0.19 / -1.30 / -0.65 | -0.16 / -1.67 / -1.82 | +0.30 / -0.00 / -1.37 | +0.61 / -2.43 / -0.61 | +0.05 / -1.08 / -1.29 |
| - Religion | -1.52 / +0.45 / -0.30 | -0.12 / -0.18 / +0.30 | -0.76 / -1.52 / -0.61 | +0.15 / -0.00 / +0.61 | -0.15 / -0.76 / -0.00 | -0.48 / -0.40 / -0.00 |
| *Random Removals ($\Delta$ from Reference)* | | | | | | |
| -3 Fields | -0.61 / +0.30 / -1.52 | +0.32 / -0.56 / -0.33 | -0.61 / -2.12 / -1.97 | -1.67 / -3.04 / -0.15 | -1.21 / -2.58 / -0.00 | -0.75 / -1.60 / -0.80 |
| -5 Fields | +1.67 / -1.22 / -3.19 | -0.54 / -0.88 / -0.96 | -0.61 / -2.58 / -2.43 | -0.30 / -0.91 / -1.37 | -1.67 / -3.95 / -1.98 | -0.29 / -1.91 / -1.99 |
| -7 Fields | -1.22 / -1.83 / -4.10 | -0.34 / -1.91 / -0.99 | +1.82 / +0.46 / -1.82 | -1.06 / -3.49 / -1.52 | -3.34 / -6.38 / -2.58 | -0.83 / -2.63 / -2.21 |

Values show $\Delta$ from baseline (exact/$\pm 1$/$\pm 2$). *green = improvement ($\Delta > +0.3$), orange = minor drop, red = significant drop ($\Delta \leq -1.5$).*

Table 5: **Contribution of Personality Components to Prediction Accuracy**: We evaluate how different personality measures (Big Five traits, 15 facets, and Locus of Control) predict responses to public health messages. Our ablation study reveals that: (1) the 15 personality facets combined with Locus of Control yield highest accuracy (77.84% average at $\pm 2$ tolerance), (2) full personality profiles marginally underperform this configuration (77.71%), and (3) individual traits show varying predictive power, with facets being most informative. Results demonstrate personality's strong predictive value while highlighting the particular importance of facet-level traits. Accuracy reported as exact/$\pm 1$/$\pm 2$ matches."

| Comparison | Components | Concern | Emotion | Harm | Motivation | Openness | Avg. |
|---|---|---|---|---|---|---|---|
| **Reference Model** | Full Personality + LOC | 30.09 / 62.01 / 83.74 | 24.51 / 45.99 / 63.79 | 44.38 / 63.37 / 79.94 | 29.64 / 59.42 / 80.40 | 33.89 / 58.81 / 80.70 | 32.50 / 57.92 / 77.71 |
| *Component Contributions ($\Delta$ from Reference)* | | | | | | | |
| - LOC | Full Personality | +1.22 / -0.61 / -0.76 | -0.59 / -0.57 / -2.76 | -0.31 / -0.30 / -0.00 | +1.36 / -0.45 / -1.22 | -0.30 / +1.68 / -1.67 | +0.28 / -0.05 / -1.28 |
| - Big5 | 15 Facets + LOC | -0.30 / +0.00 / -0.31 | -0.53 / +0.59 / +0.63 | -0.16 / +0.00 / +0.15 | -0.92 / -1.21 / -0.00 | -2.58 / +1.22 / +0.15 | -0.90 / +0.12 / +0.13 |
| - Both | Only 15 Facets | +0.91 / -1.68 / -0.00 | -0.90 / -0.80 / -1.93 | -0.46 / -0.45 / -0.00 | +0.91 / -1.06 / -0.46 | -1.06 / +0.31 / -1.37 | -0.12 / -0.74 / -0.75 |
| *Individual Component Performance ($\Delta$ from Reference)* | | | | | | | |
| | Only Big5 | -1.15 / -5.78 / -3.34 | -2.34 / -3.43 / -5.43 | -0.31 / -0.30 / -0.00 | -1.22 / -5.01 / -4.72 | -4.25 / -4.87 / -4.10 | -1.62 / -3.51 / -3.52 |
| | Only LOC | -3.04 / -7.30 / -2.28 | -4.21 / -4.12 / -5.55 | -0.16 / -0.15 / -0.00 | -6.24 / -8.20 / -1.68 | -4.86 / -5.01 / -6.23 | -3.70 / -4.96 / -3.14 |
| *Random Ablations ($\Delta$ from Reference)* | | | | | | | |
| -3 Traits | Random subset | +0.32 / -0.55 / -1.22 | -0.69 / -0.80 / -2.08 | -0.16 / -0.15 / -0.00 | +1.06 / -1.37 / -1.22 | +0.46 / +0.61 / -1.52 | +0.32 / -0.46 / -1.20 |
| -5 Traits | Random subset | +1.41 / -0.92 / -0.76 | -0.31 / -1.11 / -2.25 | -0.46 / -0.45 / -0.00 | +0.60 / -0.30 / -0.61 | -1.06 / +0.76 / -0.91 | +0.12 / -0.40 / -0.90 |
| -7 Traits | Random subset | +0.61 / -1.83 / -0.91 | -1.30 / -0.95 / -2.08 | -0.31 / -0.30 / -0.00 | +1.67 / -0.30 / -0.92 | -0.91 / -0.45 / -1.06 | -0.05 / -0.77 / -0.99 |

Values show $\Delta$ from reference model (exact/$\pm 1$/$\pm 2$). *green = improvement, orange = moderate drop (0-3%), red = large drop ($> 3\%$). LOC = Locus of Control.*

the missing piece, achieving 78.13% at exact and over 95% with $\pm 2$ tolerance. When we provide the model with only the initial opinions (prior to the stimuli), our performance drops to 32.53 at exact and 80.06% with $\pm 2$. The free form responses are also not sufficient and cause a significant drop in performance if they are the only available feature (29.96% at exact and 69.48 with $\pm 2$). Since the in-context responses to the stimuli are most indicative of one's response, we mask 3-7 of them for the random masking experiment.

### 4.3.4 OPINION PREDICTION TO VISUAL STIMULI

Our experiments in Table 7 reveal a nuanced relationship between visual stimuli and opinion prediction: Images contribute critically to exact-match accuracy, with removal causing a 36.32% drop in motivation and 30.85% in concern. Average exact-match accuracy falls from $77.58 \rightarrow 57.25\%$ (-20.33), proving visuals to be strong affective anchors for categorical judgments. In minimal-input conditions (Personality + Demographics + Locus of Control, image removal improves $\pm 1, 2$ accuracy $(+5.46/+3.80)$, implying that the textual signal dominates nuanced opinion spectra. This supports a dual-process model of AI opinion prediction: System 1 (Fast): Image-driven, affective processing dominates initial classification. System 2 (Slow): Deliberative analysis of stable traits (P+D+LOC) enables fine-grained prediction.

Table 6: **Impact of In-Context Responses on Prediction Accuracy**: We evaluate how different response components (initial opinions, free-form text, and structured Q/A responses) contribute to predicting public health message reception. Using Llama 11B, we demonstrate that structured Q/A responses alone achieve comparable performance ($\Delta + 0.68 / + 0.10 / + 0.16$) to the full model, while initial opinions and free-form text show limited predictive value individually ($\Delta - 44.32 / - 26.61 / - 15.66$ and -47.49/-38.63/-26.24 respectively). Removing all response components causes severe performance degradation (-44.32 average exact match), highlighting their critical role in accurate prediction. Results are reported as exact/±1/±2 match accuracy across five psychological dimensions.

| Comparison | Components | Concern | Emotion | Harm | Motivation | Openness | Avg. |
|---|---|---|---|---|---|---|---|
| **Reference Model** | All components | 98.63 / 99.70 / 100.00 | 99.71 / 99.83 / 99.89 | 49.54 / 76.29 / 92.10 | 98.78 / 98.94 / 99.09 | 40.58 / 73.10 / 87.54 | 77.45 / 89.57 / 95.72 |
| *Component Contributions (Δ from Reference)* | | | | | | | |
| - Initial Opinions | All except initial | +0.15 / -0.31 / -0.15 | +0.00 / +0.00 / +0.00 | +2.13 / +3.04 / +1.21 | -0.76 / -0.31 / -0.15 | +2.13 / -2.13 / -0.76 | +0.73 / +0.06 / +0.03 |
| - Free Form | All except free form | +1.07 / +0.30 / +0.00 | -0.01 / -0.04 / -0.04 | +1.37 / +2.59 / +1.67 | -0.30 / -0.16 / -0.15 | +0.76 / +0.15 / -0.15 | +0.57 / +0.57 / +0.27 |
| - Responses | Only responses | +1.07 / +0.00 / +0.00 | +0.06 / +0.04 / +0.01 | +1.98 / +3.19 / +1.37 | -1.67 / -0.76 / -0.46 | +1.97 / -1.98 / -0.15 | +0.68 / +0.10 / +0.16 |
| *Individual Component Performance (Δ from Reference)* | | | | | | | |
| | Only initial | -64.74 / -29.34 / -13.07 | -78.15 / -55.71 / -41.36 | -5.92 / -8.81 / -2.59 | -66.71 / -33.89 / -17.48 | -6.08 / -5.32 / -3.80 | -44.32 / -26.61 / -15.66 |
| | Only free form | -72.03 / -47.12 / -25.38 | -79.41 / -59.86 / -45.97 | -5.32 / -12.92 / -11.55 | -70.21 / -49.70 / -28.42 | -10.49 / -23.56 / -19.91 | -47.49 / -38.63 / -26.24 |
| *Random Ablations (Δ from Reference)* | | | | | | | |
| -3 Fields | Random subset | -16.72 / -6.84 / -2.58 | -17.21 / -8.96 / -4.64 | -0.45 / +1.37 / +1.37 | -21.27 / -11.86 / -6.08 | -2.13 / +19.76 / -1.37 | -11.56 / -6.14 / -2.66 |
| -5 Fields | Random subset | -27.96 / -12.92 / -5.17 | -29.54 / -15.46 / -8.43 | -2.43 / -1.67 / +0.76 | -33.89 / -19.15 / -8.97 | -3.04 / -5.93 / -2.28 | -19.37 / -11.02 / -4.81 |
| -7 Fields | Random subset | -39.21 / -19.91 / -7.90 | -40.93 / -23.04 / -13.04 | -0.91 / -3.04 / -3.35 | -45.44 / -27.06 / -12.77 | -7.75 / -8.21 / -3.50 | -26.85 / -16.25 / -8.11 |

*Values show Δ from reference model (exact/±1/±2). Color coding: green = improvement, orange = moderate drop (0-10%), red = large drop (> 10%).*

Table 7: **Impact of Visual and Contextual Inputs on Opinion Prediction**

| Comparison | Inputs Used | Concern | Emotion | Harm | Motivation | Openness | Avg. |
|---|---|---|---|---|---|---|---|
| *Visual Impact Analysis (Image Removal)* | | | | | | | |
| | All fields (w/ image) | 98.63 / 99.54 / 100.00 | 99.73 / 99.83 / 99.89 | 49.24 / 74.32 / 90.58 | 99.09 / 99.24 / 99.39 | 41.19 / 71.73 / 87.08 | 77.58 / 88.93 / 95.39 |
| | All fields - image | 67.78 / 88.75 / 95.90 | 80.64 / 92.55 / 95.71 | 40.88 / 68.09 / 93.92 | 62.77 / 85.56 / 94.07 | 34.19 / 69.00 / 84.65 | 57.25 / 80.79 / 92.85 |
| **Change** | **Image contribution** | -30.85 / -10.79 / -4.10 | -19.09 / -7.28 / -4.18 | -8.36 / -6.23 / +3.34 | -36.32 / -13.68 / -5.32 | -7.00 / -2.73 / -2.43 | -20.33 / -8.14 / -2.54 |
| *Contextual Impact Analysis (Responses + Image Removal)* | | | | | | | |
| | P+D+LOC (w/ image) | 35.41 / 63.98 / 84.35 | 25.96 / 47.23 / 63.69 | 42.71 / 62.46 / 81.46 | 31.31 / 62.77 / 79.64 | 32.07 / 61.40 / 81.46 | 33.49 / 59.57 / 78.12 |
| | P+D+LOC - both | 36.17 / 72.19 / 86.63 | 23.94 / 52.83 / 65.35 | 39.06 / 65.05 / 92.25 | 30.85 / 67.93 / 81.91 | 33.28 / 67.17 / 83.43 | 32.66 / 65.03 / 81.92 |
| **Change** | **Combined effect** | +0.76 / +8.21 / +2.28 | -2.02 / +5.60 / +1.66 | -3.65 / +2.59 / +10.79 | -0.46 / +5.16 / +2.27 | +1.21 / +5.77 / +1.97 | -0.83 / +5.46 / +3.80 |

*LOC = Locus of Control. green = improvement, orange = moderate drop (0-10%), red = large drop (> 20%).*

## 5 CONCLUSION

This paper introduces PHORECAST, a novel multi-modal dataset that enables modeling of human reactions to public health campaigns using demographics and psychographic factors and personality traits. With training using PHORECAST, we capture fine-grained individual traits, inclusive of demographics, personality, and contextual behaviors, and demonstrate the utility of this dataset in predicting responses to health messaging. Our results highlight both the shortcomings of existing LLM/VLM models and the promise of individualized reasoning in shaping better human-aligned public health interventions. PHORECAST provides a foundation for future research in human-centric AI systems that aim to navigate the complexity and diversity of real-world decision-making.

**Discussion, Limitations and Future Directions:** This works present several promising avenues for future research. First, this dataset comprises primarily of English-speaking participants, which may bias the findings and limit global applicability. Cultural, linguistic, and geopolitical factors critically shape attitudes towards public health issues, and future efforts must broaden the participant base to support more globally representative insights. Second, while our models incorporate rich contextual features, they remain static with respect to time. Health beliefs are not fixed; they evolve with social, political, and personal contexts. Modeling temporal dynamics is essential to anticipate these shifts and to evaluate the long-term impact of interventions.

PHORECAST introduces many possibilities to more responsive, adaptive, and pluralistic public health communication strategies. In addition to addressing the issues raised above, we envision future work that explores hybrid modeling techniques, such as mixture-of-experts or continual learning, that balance individual expressivity with societal objectives, forging a path toward more ethical, inclusive, and effective public health communication.

**Contraindications** PHORECAST is not a global dataset. As such, PHORECAST should not be used to infer cross-cultural, multilingual, or non-U.S. population responses to health communication. Researchers seeking to generalize beyond U.S.-based, English-speaking populations should treat PHORECAST results as hypothesis-generating only, and pursue follow-up studies with more representative data.

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
