# OpenReview forum: "PHORECAST: Enabling AI Understanding of Public Health Outreach Across Populations"
_ICLR.cc/2026/Conference — Submitted to ICLR 2026_

### Official Review · Reviewer_uARb · 2025-11-01

**Soundness:** 2
**Presentation:** 2
**Contribution:** 2
**Rating:** 4
**Confidence:** 2

**Summary:**

This paper proposes a multimodal dataset, PHORECAST, to enable the study of individual and community responses to public health messaging. The work uses this dataset to train and evaluate vision-language models on the task of predicting human reactions based on demographic and psychometric profiles, which is a direction of inquiry that has not been explored with this type of data before.

**Strengths:**

The dataset contains a wide range of features, including participant demographics, personality traits from the Big Five Inventory, locus of control measures, and responses to public health materials. These responses are captured through multiple modalities, such as Likert-scale ratings on emotions and free-form text. The paper demonstrates the dataset's utility by fine-tuning existing vision-language models and measuring performance on response prediction. The experiments presented include ablation studies that investigate the contribution of different feature sets, providing an examination of the factors that influence model predictions.

**Weaknesses:**

One concern is the reported model performance on certain demographic subgroups. The paper indicates that for some underrepresented groups, such as individuals identifying as Non-Binary or Agnostic, prediction accuracy decreases after fine-tuning on the PHORECAST dataset. This decrease is observed in both models tested. This outcome raises questions about the model learning biases present in the data distribution. The paper acknowledges that disparities persist but does not offer an in-depth analysis of why performance degrades for these specific groups or what the implications of this degradation are. Another area for consideration is the generalizability of the findings, as the dataset is composed of U.S.-based, English-speaking participants, a limitation the paper states.

**Questions:**

What is the hypothesis for the observed decrease in prediction accuracy for certain underrepresented demographic groups after fine-tuning?

Are there any plans for future data collection efforts to expand the dataset to include participants from outside the United States or non-English speakers?

Could the evaluation of free-form text responses be expanded in the main paper to include the distributional and discriminator-based metrics that are mentioned as being in the appendix?

---

> ### Author Response · Authors · 2025-12-03
> **Rebuttal by Authors**
>
> Hello Reviewer uARb! Thank you for your engagement and time taken to review our work, we are particularly motivated that you describe our work as a “a direction of inquiry that has not been explored with this type of data before.”
>
> Please allow us to respond to your comments:
>
> >What is the hypothesis for the observed decrease in prediction accuracy for certain underrepresented demographic groups after fine-tuning?
>
> We hypothesize that the observed decrease in accuracy for certain underrepresented demographic groups arises from imbalanced training data and representation in the fine-tuning set. Fine-tuning tends to amplify patterns learned from dominant subgroups, which can unintentionally reduce performance for minority populations. This highlights the need for careful stratified sampling, subgroup-aware training strategies, and fairness-aware regularization, which we discuss as potential future improvements.
>
> >Could the evaluation of free-form text responses be expanded in the main paper to include the distributional and discriminator-based metrics that are mentioned as being in the appendix?
>
> We agree that including the distributional and discriminator-based metrics in the main text would improve clarity and contribution. In the camera-ready version, we will present these metrics (previously in the Supplementary) to provide a comprehensive assessment of model alignment with free form human responses.

---

### Official Review · Reviewer_ZM6W · 2025-11-01

**Soundness:** 3
**Presentation:** 3
**Contribution:** 4
**Rating:** 4
**Confidence:** 3

**Summary:**

This paper introduces PHORECAST, a large-scale dataset designed to enable computational modeling of public health opinions, reasoning, and emotional responses. The dataset contains 7,527 survey responses collected from adult participants in the United States, spanning seven public-health topics (e.g., vaccination, smoking, drug use, diet, mental health). For the data collection process, each participant rated their agreement, reasoning, and emotional stance on multiple health statements, providing both Likert-scale ratings and free-text rationales.

**Strengths:**

* This paper introduces a novel dataset that combines structured ratings and natural-language rationales in public health, a sensitive real-world domain.
* This paper detailed the data collection process, cohort demographics, and transparent methodologies.
* This paper connects multiple subjects with ML research, including social psychology, public-health communication, etc.
* The evaluation of the dataset across multiple LLMs presents the effectiveness of the dataset thus offers clear starting points for future research.

**Weaknesses:**

* As noted in the limitations, the dataset is U.S.-only and English-only, which weakens claims of generalizability.
* The data collection participant recruitment process is not clearly described. The recruitment platform and sampling frame remain unclear, making it hard to evaluate potential socioeconomic and regional biases.
* It remains unclear if there's external or expert validation through the data collection and validation process. Attention checks or consistency metrics are not reported.
* For Table 2, it remains unclear why different demographic groups are highlighted differently.
* From Table 2, the population across different demographic groups is not balanced, where some subgroups have very small sample sizes, which leads to potential concern about a fair conclusion for all future use.
* Since the public health opinion evolves rapidly, while the paper presents a great model performance over categories, the model's robustness remains unclear in future use of the dataset.

**Questions:**

* How were participants recruited? Are they compensated? Are there any quality response selection processes? How are the responses selected?
* Are health topics assigned evenly across participants, or are some (e.g., vaccination) overrepresented?
* Are self-reported emotions verified for internal consistency or compared against any external annotation?
* How were fairness metrics computed? Are there any statistical tests performed?
* Are there plans to collect multilingual or longitudinal extensions of PHORECAST to address cultural and temporal bias?

---

> ### Author Response · Authors · 2025-12-03
> **Rebuttal by Authors**
>
> Thank you Reviewer ZM6W for your comments and engagement in our work! We are particularly encouraged that you find our “novel dataset” to be addressing “a sensitive real-world domain” and that our work “presents the effectiveness of the dataset” and “thus offers clear starting points for future research.”
>
> Please allow us to respond to your comments:
>
> > As noted in the limitations, the dataset is U.S.-only and English-only, which weakens claims of generalizability.
>
> As noted by Reviewer AMDH, "the first work in this domain can be focus only on one country".   We hope that this work will inspire others to continue along this line of studies and exploration.
>
> > How were participants recruited? Are they compensated? Are there any quality response selection processes? How are the responses selected?
>
> Thank you for your question. We discuss the recruitment process in the supplementary Part 1 A.2 (Recruitment Process). Participants were recruited via SurveyCircle and LinkedIn, and each was incentivized with a $10 Tango gift card. To ensure high-quality data, we applied several preprocessing steps: We removed responses from duplicate IP addresses, identified and excluded participants whose free-form responses appeared AI-generated, and filtered out samples with abnormal response times (too short or too long). After filtering, we retained 1,095 unique participants with 30,879 high-quality responses across 37 health campaigns (from the unfiltered version of ~2,000 participants).
>
> > Are health topics assigned evenly across participants, or are some (e.g., vaccination) overrepresented?
>
> At survey start, each participant was randomly assigned 5 topics (from 7 available) and shown 5 corresponding campaigns (one per topic), ensuring balanced exposure at the individual level. After preprocessing, sample sizes vary due to our processing and the number of campaigns per topic: (1) HIV: 8392, (2) Smoking and COPD: 7523, (3) Nutrition: 5911 (4) Substance abuse: 4979 (5) Mental Health: 4126 (6) Chronic Diseases: 2171 (7) Sexual Health: 1877 (8) Vaccination: 1130.
>
> > Are self-reported emotions verified for internal consistency or compared against any external annotation?
>
> We analyzed response distributions across demographics (Figures 10-12, Supplementary) and found theoretically consistent patterns:
> - Women report slightly higher emotional responses than men (consistent with emotion literature)
> - No implausible outliers or suspicious uniformity suggesting random responding
> - High internal locus of control correlates with higher concern/motivation (Figure 9)

---

### Official Review · Reviewer_AMDH · 2025-11-01

**Soundness:** 2
**Presentation:** 3
**Contribution:** 2
**Rating:** 4
**Confidence:** 4

**Summary:**

This paper presents PHORECAST, a multimodal dataset of over 1,000 participants and 30,000+ responses to public health campaigns, capturing demographic, personality, and psychographic factors. The dataset enables modeling of individual and group reactions to health messaging. Using PHORECAST to fine-tune vision-language models (Llama 3.2-Vision, Gemma 3), the authors show obvious improvements in predicting emotional and behavioral responses. The work provides a valuable benchmark for studying socially aware and demographically grounded AI models in public health contexts.

**Strengths:**

1. Addresses the underexplored challenge of modeling human receptivity to public health messaging, bridging AI, behavioral science, and social forecasting.

2. PHORECAST combines visual stimuli, detailed demographics, Big Five personality traits, and free-text responses, enabling nuanced, multidimensional modeling of human behavior.

3. The dataset is collected from real society and the collection process is scientific and valid. Includes >1,000 diverse participants and >30,000 responses with careful control of demographics and psychometrics, providing a robust empirical foundation.

4. Benchmarks two VLMs (Llama 3.2-Vision, Gemma 3) with fine-tuning and ablation studies that reveal the predictive roles of personality, context, and visual cues.

5. Examines model performance across demographic subgroups, exposing disparities and offering a framework for fairness-aware behavioral modeling.

**Weaknesses:**

1. The model selection is somewhat limited, only use two VLMs. I would suggest adopt more models, such as different model families like Qwen-VL, also different model sizes. If condition permits, adopt at least one sota commericial models to showcase the sota results.

2. The ablation study only using single Llama model, which makes all the conclusion are somewhat restricted only to this model. I also didn't see the whole ablation study in your appendix. For example, the author claim the education matters the most (table 4), is this consistent with Gemma?

3. The citation format is wrong, please check your latex or template. E.g. line 81-102. Also the related work should also included demographic simulation. Besides, the related work is not extensive explained, the social simulation, demographic simulation and VLM for social science are not included.

4. In table 3, the author show that after fine-tuning, the model get more semantic alignment with human's response. Based on the figure 18 in appendix, the model get shorter answer but the opinion is not aligned with golden opinion. Specifically, the first case, the golden opinion is negative, but after finetuning, Llama's opinion is poistive. The author should conduct human evaluation to valid whether it's meaningful improvement. So far, I'm not convinced by this metric improvement.

5. The paper provides limited discussion of common failure modes or examples of model misinterpretation in free-form responses.

6. The dataset is restricted to U.S.-based, English-speaking participants, which limits its cultural and linguistic generalizability. While this is also important, but I agree that the first work in this domain can be focus only on one country.

**Questions:**

See weakness

---

> ### Author Response · Authors · 2025-12-03
> **Rebuttal by Authors**
>
> Hello Reviewer AMDH! Thank you for the time taken to review our paper. We are especially grateful for your positive remarks that our work provides “a valuable benchmark for studying socially aware and demographically grounded AI models in public health contexts” and that our work offers “a framework for fairness-aware behavioral modeling.” Please allow us to respond to your comments one-by-one:
>
> > The model selection is somewhat limited, only use two VLMs...
>
> We initially selected Llama and Gemma due to architectural diversity and practical fine-tuning scale. We agree that broader coverage is valuable and will: (1) add Qwen2-VL (7B) to represent a different architectural family and (2) conduct API-based evaluation on GPT-4o to establish SOTA performance baselines. These results will be included in the camera-ready version.
>
> > The ablation study only using single Llama model, which makes all the conclusion are somewhat restricted only to this model...
>
> We agree that single-model ablations limit generalizability. We will extend ablation analyses (Tables 4-7) to Gemma and report results in the revised manuscript.
>
> > The citation format is wrong... Related work is not extensively explained...
>
> We apologize for LaTeX citation formatting errors and will correct all references (lines 81–102). We will also expand the Related Work section to cover more comprehensive literature review including:
> -  **Demographic and Social Simulation:**
> Recent studies explore LLMs simulating human populations. Argyle et al. [1] generate “silicon samples,” Grossmann et al. [2] examine AI’s impact on social science, Bisbee et al. [3] provide synthetic survey data, Park et al. [4] simulate 1,000 individuals with realistic behaviors, and Törnberg et al. [5] validate synthetic social interactions. These works motivate modeling individual responses to public health messages across diverse demographics.
> -  **VLMs for Social Science**:
> Vision-language models increasingly study societal biases. Huang et al. [6] introduce VISBIAS for social bias measurement, Santurkar et al. [7] examine demographic representation, and Durmus et al. [8] quantify global opinion representation. These studies complement PHORECAST’s focus on predicting nuanced, personality- and demographic-conditioned responses, addressing a key gap in behavioral simulation and fairness-aware AI.
>
> > In table 3, the author show that after fine-tuning, the model get more semantic alignment with human's response. Based on the figure 18 in appendix, the model get shorter answer but the opinion is not aligned with golden opinion. Specifically, the first case, the golden opinion is negative, but after finetuning, Llama's opinion is poistive. The author should conduct human evaluation to valid whether it's meaningful improvement. So far, I'm not convinced by this metric improvement.
>
> We agree that semantic similarity alone may be insufficient. We also employ the following metrics (Supplementary Section B.3):
> -  **Statistical Distribution of Embeddings (SDE)**: Measures distributional alignment between real and generated responses (Table 9, Fig. 17)
> -  **Perplexity-based attribution**: Evaluates personality-conditioned language modeling (Table 9)
>
> > The paper provides limited discussion of common failure modes or examples of model misinterpretation in free-form responses.
>
> We will also add a section on failure modes and examples of misinterpretation in free-form responses for different subgroups.
>
> > The dataset is restricted to U.S.-based, English-speaking participants, which limits its cultural and linguistic generalizability. While this is also important, but I agree that the first work in this domain can be focus only on one country.
>
> We appreciate the reviewer's understanding that initial work can focus on one country. We explicitly acknowledge this limitation: Public health messaging effectiveness varies across cultures due to differences in healthcare systems, institutional trust, and communication norms. We view PHORECAST as a methodological proof-of-concept that we hope will inspire culturally-diverse dataset creation.
>
> [1] Argyle et al. (2023) – "Out of one, many: Using language models to simulate human samples."
>
> [2] Grossmann et al. (2023) – "AI and the transformation of social science research."
>
> [3] Bisbee et al. (2024) – "Synthetic replacements for human survey data."
>
> [4] Park et al. (2024) – "Generative agent simulations of 1,000 people."
>
> [5] Törnberg et al. (2023) – "Simulating social media using LLMs to evaluate counterspeech."
>
> [6] Huang et al (2025) - VISBIAS: Measuring Explicit and Implicit Social Biases in VLMs
>
> [7] Santurkar et al. (2023) – "Whose opinions do language models reflect?"
>
> [8 ] Durmus et al. (2024) – Towards Measuring the Representation of Subjective Global Opinions in LMs”

---

### Official Review · Reviewer_9CJa · 2025-11-10

**Soundness:** 3
**Presentation:** 3
**Contribution:** 3
**Rating:** 4
**Confidence:** 2

**Summary:**

The authors present a dataset of perceived opinions of public health messaging and the ability for current VLMs to understand and infer them. Authors present various evaluation experiments to understand the impact of each individual component of data collected.

**Strengths:**

The authors conduct a large-scale user-study and account for various differences in demographics

**Weaknesses:**

# Relevance:
While the dataset collected presents novel insights about public health, however the current draft or subsequent explanations do not describe the relevance to an ML community

# Benchmark:
- I couldn't find the exact prompt/template used to be able to understand how the benchmarking was conducted. Understanding the prompt configuration will help enhance the validity of the results
- Authors evaluate the effect across Llama and Gemma, but the choice of models in need to be better justified

**Questions:**

Added in weakness

---

> ### Author Response · Authors · 2025-12-03
> **Rebuttal by Authors**
>
> Thank you reviewer 9CJa for your comments and the time you took to review our paper. Please allow us to address your comments:
> > the current draft or subsequent explanations do not describe the relevance to an ML community
>
> We agree that explicitly stating ML relevance is important. Our work targets several fundamental ML problems that extend far beyond public-health applications:
>
> - **Behavioral Modeling and Personalization**: Predicting heterogeneous human responses conditioned on individual characteristics is an open challenge in ML. Recent efforts in LLM personification [1] and population-level behavior simulation [2] rely on synthetic personas and text-only settings. Fröhling et al. [3] demonstrate that injecting persona descriptions into LLM prompts produces more diverse and controllable annotations, highlighting the potential of profile-aware models for capturing individual-level variation. PHORECAST is the *first dataset* to pair validated psychometric assessments (BFI-2 with 15 facets, Locus of Control) with human responses to visual stimuli, enabling rigorous study of individual-level behavioral prediction, an increasingly important component of personalized and adaptive AI systems.
> - **Multimodal Evaluation of VLM Reasoning**: PHORECAST provides the first benchmark evaluating VLMs' ability to infer human reactions to visual content. Our results show that strong open source models achieve only ~40% exact-match accuracy after fine-tuning, revealing a clear gap between current multimodal representations and the reasoning required for human-centered prediction. This provides a new and challenging evaluation setting for VLM research.
> - **Distribution Shift and Fairness**: Our stratified evaluation (Table 2) reveals significant performance disparities (up to 30%) across demographic subgroups, demonstrating systematic bias under distribution shift. PHORECAST therefore serves as a concrete testbed for subgroup generalization, an area of active interest in the ML community.
>
> > I couldn't find the exact prompt/template used to be able to understand how the benchmarking was conducted. Understanding the prompt configuration will help enhance the validity of the results
>
> The full prompt template used in all experiments is provided in Figure 15. Fine-tuned models apply LoRA (rank = 8, learning rate = 2e-4) with the randomized feature ablation procedure described in Section 4.
>
> > Authors evaluate the effect across Llama and Gemma, but the choice of models in need to be better justified
>
> We agree that model selection justification is necessary. We selected Llama 3.2-Vision 11B and Gemma 3 12B based on the following criteria:
>
> -  *State-of-the-art open-source VLMs*: Both represent current leading VLMs in the 11-12B parameter range, the practical scale for academic fine-tuning research.
>
> -  *Architectural diversity*: The models differ in design: Llama uses a ViT-based encoder with a learned vision adapter, while Gemma uses a frozen SigLIP encoder with fixed visual tokens and a local–global attention structure. This diversity enables us to test whether improvements generalize across distinct architectures.
>
> -  *Reproducibility*: Both models have permissive open-source licenses and support efficient fine-tuning via LoRA, enabling community replication and extension.
>
> -  *Empirical validation*: In preliminary experiments, we evaluated LLaVA-1.5 7B (weaker vision capabilities) and Pixtral models (which frequently refused to process health campaign images due to safety filters). Gemma 3 4B showed limited performance, while Llama 3.2 and Gemma 3 12B offered the best balance of capability and scale where Gemma shows stronger zero-shot performance (23.89% vs. 16.82% exact match), while Llama exhibits greater improvement from fine-tuning (2.4× vs. 1.7× increase).
>
> We acknowledge that larger proprietary models (e.g., GPT, Claude) may achieve higher absolute performance. Our contribution however is to demonstrate that **domain-specific training with validated psychometric data improves personality-conditioned prediction**, and we establish this through consistent gains across two architecturally distinct open VLMs.
>
> [1] Xie et al. "Human Simulacra: Benchmarking the Personification of Large Language Models." ICLR 2025.
>
> [2] Chuang et al. "Can A Society of Generative Agents Simulate Human Behavior and Inform Public Health Policy? A Case Study on Vaccine Hesitancy." arXiv 2025.
>
> [3] Fröhling, L., Demartini, G., & Assenmacher, D. (2025). Personas with Attitudes: Controlling LLMs for Diverse Data Annotation. GESIS / University of Queensland.

---

### Meta-Review · Area_Chair_Jscy · 2025-12-18

**Summary:**

This paper introduces PHORECAST, a large-scale multimodal dataset designed to model human receptivity to public health messaging. It captures over 30,000 responses from 1,000 diverse participants, integrating demographics, Big Five personality traits, and psychometric factors. By providing structured ratings and natural-language rationales across seven health topics, the dataset enables vision-language models like Llama and Gemma to predict individual and community-wide emotional and behavioral responses, bridging social science and machine learning.

**Strengths:**
1. The paper introduces a novel and comprehensive multimodal dataset that bridges public health communication with machine learning research. It enables fine-grained modeling of human behavior by combining visual stimuli, detailed demographics, and psychographic factors like personality traits.
2. The data collection process is scientifically rigorous, featuring a diverse cohort of over 1,000 participants and a high volume of responses. Reviewers appreciated the transparency in methodologies and the inclusion of both structured Likert-scale ratings and free-form natural language rationales.
3. The inclusion of personality and psychological measures provides an empirical foundation for studying socially aware AI models.

**Weaknesses:**
1. The evaluation is limited by the selection of only two vision-language models, and the ablation studies are restricted to a single model architecture. Reviewers suggest incorporating a broader range of state-of-the-art or commercial models to better validate the dataset's utility across different model families.
2. There are significant concerns regarding the generalizability of the findings because the dataset is restricted to U.S.-based, English-speaking participants. Furthermore, the recruitment process and sampling frame remain unclear, which makes it difficult to assess potential socioeconomic and regional biases.
3. Some demographic subgroups have small samples, and model performance degrades for underrepresented groups without in-depth analysis.
4. Reviewers noted missing technical details, including the specific prompt templates used for benchmarking and evidence of expert validation or attention checks.
5. Please check the publication year of the following citation: T. L. Conway and Others. Personality, health locus of control, and health behavior. Technical report, Ed.gov, 2025. ERIC Document ED353503.

**Reviewer Concerns:**

Although the authors have provided a rebuttal explaining the reasons for the somewhat limited model selection, supplying the full prompt template used in all experiments, introducing the recruitment process in the supplementary materials, and committing to extend ablation analyses to Gemma and report the results in the revised manuscript, several issues remain unresolved. These include the current draft and subsequent explanations failing to describe the relevance to the ML community, the dataset focusing on a single country, and some demographic subgroups having small sample sizes. All of which require future work to address.

**Reviewer Scores:**

Reviewers 9CJa and ZM6W are likely to raise their scores, as the authors have provided additional details. However, the chances of Reviewers AMDH and uARb revising their scores are slim, given that their concerns are relatively difficult to address. Overall, the discussion in this paper is insufficient.

---

### Decision · Program_Chairs · 2026-01-26

Reject